health and disease and epidemiology/differential equations

epidemic, SEIR model, concave function, difference equations, Pareto optimization

**Author for correspondence:**
R. C. Tyson
e-mail: rebecca.tyson@ubc.ca

# Optimal shutdown strategies for COVID-19 with economic and mortality costs: British Columbia as a case study

M. T. Barlow[1], N. D. Marshall[2] and R. C. Tyson[3]

[1]Department of Mathematics, University of British Columbia, Vancouver, British Columbia, Canada V6T 1Z2
[2]Department of Mathematics and Statistics, McGill University, 805 Sherbrooke Street West, Montreal, Quebec, Canada H3A 0B9
[3]CMPS Department, University of British Columbia Okanagan, 1177 Research Road, Kelowna, British Columbia, Canada V1V 1V7

RCT, 0000-0002-7373-4473

Decision makers with the responsibility of managing policy for the COVID-19 epidemic have faced difficult choices in balancing the competing claims of saving lives and the high economic cost of shutdowns. In this paper, we formulate a model with both epidemiological and economic content to assist this decision-making process. We consider two ways to handle the balance between economic costs and deaths. First, we use the statistical value of life, which in Canada is about C$7 million, to optimize over a single variable, which is the sum of the economic cost and the value of lives lost. Our second method is to calculate the Pareto optimal front when we look at the two variables—deaths and economic costs. In both cases we find that, for most parameter values, the optimal policy is to adopt an initial shutdown level which reduces the reproduction number of the epidemic to close to 1. This level is then reduced once a vaccination programme is underway. Our model also indicates that an oscillating policy of strict and mild shutdowns is less effective than a policy which maintains a moderate shutdown level.

## 1. Introduction

The COVID-19 pandemic led many countries to introduce an extensive economic and social lockdown to limit the spread of the disease [1]. While these measures are very expensive [2], in most regions where the measures were applied relatively early and with sufficient stringency, they were successful in stopping the growth of the epidemic [3]. After an initial period of shutdown, many jurisdictions have partially reopened their

economies, and in some cases this has led to a second surge of infections [4] including in Canada [5], reopening the question of whether or not to impose another lockdown.

Some kind of balance has to be struck between saving lives and the economic (and social) cost of the lockdown [6,7]. This paper presents a simple model, with both economic and epidemiological content, to help assess the options. In particular, we aim to determine what type of shutdown strategy ($X_t$) minimizes costs over the period of the epidemic. The data and numbers are for the Canadian province of British Columbia (BC). We remark that any decision on the extent of the lockdown has to take many factors into account. We have chosen to express those factors which we do consider in monetary terms; inevitably this means that some important aspects will have been left out. The optimization we perform should be viewed as information that can assist in making a complex decision, rather than as a simple prescription.

We start our model at an early stage of the epidemic, corresponding roughly to the situation in BC in March 2020; thus we consider 'what should have been done' as well as 'what should now be done'. We find that the minimum cost scenario does include a significant level of economic shutdown in order to ensure reduced transmission. In particular, we obtain a high level of shutdown at the beginning of the epidemic, which is monotonically decreased until the population has been vaccinated.

Since this project was started research on COVID-19 has progressed rapidly, and the nature of the epidemic has changed due to the evolution of new variants. We have chosen to 'freeze' our analysis at a time point in late Autumn 2020, i.e. before the importance of new variants was recognized, or information on vaccine efficacy was available.

## 2. Model

Our model contains two parts: one part describing the disease dynamics; the second describing the economic costs and optimal control. We separately present each part of the model below.

### 2.1. Disease dynamics

Our model for the epidemic is of standard compartmental type. For clarity, we split it into two parts. The first part is a standard SEIQ model, where $S$ is susceptible, $E$ is exposed, $I$ is infectious and $Q$ is quarantined (i.e. no longer mixing with the susceptible population while infectious):

$$S_{t+1} - S_t = -v_t - \beta_t I_t \frac{S_t}{N}, \tag{2.1a}$$

$$E_{t+1} - E_t = \beta_t I_i \frac{S_t}{N} - \gamma E_t \tag{2.1b}$$

$$I_{t+1} - I_t = \gamma E_t - \alpha I_t \tag{2.1c}$$

$$\text{and} \quad Q_{t+1} - Q_t = \alpha I_t + v_t. \tag{2.1d}$$

The time-dependent infectivity parameter $\beta_t$ depends on the amount of economic lockdown. $v_t$ denotes the number of vaccinations on day $t$—see below for details of how these are defined.

We initialize the model by taking

$$E_0 = I_0 = e_0,$$

that is, the initial number of COVID-19 carriers is split evenly between the exposed and infectious compartments.

To model the progress of patients through the medical system, we separate the $Q$ bin into smaller bins depending on quarantine state. These bins are named $M, H, W, U, R, D$. Here $M_t$ is the number of patients who are mildly ill (or asymptomatic) and therefore quarantining at home, $H_t$ and $U_t$ are the numbers of patients in hospital (but not ICU) and ICU beds, respectively, $R_t$ is the number of recovered patients, and $D_t$ is the number of deceased patients. These population compartments, and the connections between them, are illustrated in figure 1.

We wish to handle possible overload of the medical system, so we set the maximum capacity of hospital and ICU beds to be $H_{\max}$ and $U_{\max}$, respectively. For patients needing a hospital (non-ICU) bed, we introduce the $W_t$ compartment: the number of patients waiting for hospital beds. At time $t + 1$ the demand for hospital beds is

$$\tilde{H}_{t+1} = H_t(1 - \kappa_H) + W_t(1 - \kappa_W) + \alpha I_t p_H.$$

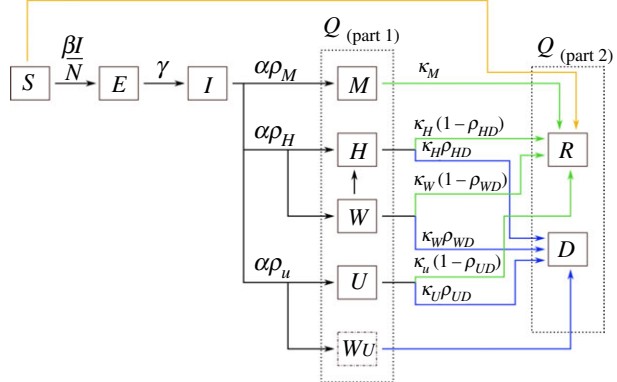

**Figure 1.** The compartments of the disease model. The two dashed line boxes surround all of those compartments that are still infectious, but are no longer transmitting disease as they are quarantined at home or in hospital (the $Q$ compartment in the model). The green arrows indicate transitions from sick states to the recovered state, while blue arrows indicate transitions from sick states to the deceased state. The orange line denotes vaccinations. The variables are fractions of the total population: $S$ = susceptible, $E$ = exposed, $I$ = infectious and not quarantined, $M$ = infected but only mildly ill, $H$ = infected and in hospital, $W$ = infected and waiting for space to open up in hospital, $U$ = infected and in the ICU, $R$ = recovered, and $D$ = deceased. The $W$ individuals are assumed to be quarantined at home. $W_U$ is a instantaneous compartment containing individuals who need ICU but cannot be admitted there; these individuals are moved immediately into the $D$ compartment.

If demand is greater than supply, we move as many patients as possible into hospital, and the remainder go into the $W$ container. Thus we set

$$H_{t+1} = \min(H_{\max}, \tilde{H}_{t+1}) \quad \text{and} \quad W_{t+1} = \max(\tilde{H}_{t+1} - H_{\max}, 0).$$

We follow a similar procedure for the ICU patients, except that (as these patients are presumably very ill) any excess is immediately moved into the $D$ container; we write

$$D_{t+1}^{EU} = \max(0, U_t(1 - \kappa_U) + \alpha I_t p_U),$$

for the number of these deaths. Thus, the model for the quarantined group is written

$$M_{t+1} = M_t(1 - \kappa_M) + \alpha p_M I_t, \tag{2.2a}$$

$$H_{t+1} = \min(H_{\max}, H_t(1 - \kappa_H) + W_t(1 - \kappa_W) + \alpha I_t p_H) \tag{2.2b}$$

$$W_{t+1} = \max(0, H_t(1 - \kappa_H) + W_t(1 - \kappa_W) + \alpha I_t p_H - H_{\max}) \tag{2.2c}$$

$$U_{t+1} = \min(U_{\max}, U_t(1 - \kappa_U) + \alpha I_t p_U) \tag{2.2d}$$

$$R_{t+1} = R_t + \kappa_M M_t + \kappa_H(1 - p_{HD})H_t + \kappa_W(1 - p_{WD})W_t + \kappa_U(1 - p_{UD})U_t + v_t \tag{2.2e}$$

$$\text{and} \quad D_{t+1} = D_t + \kappa_H p_{HD} H_t + \kappa_W p_{WD} W_t + \kappa_U p_{UD} U_t + D_{t+1}^{EU}. \tag{2.2f}$$

We have $Q_t = M_t + H_t + W_t + U_t + R_t + D_t$; note that $Q$ satisfies (2.1$d$).

We count as *excess deaths* those deaths arising from overload of the health system. This consists of two terms: the number of patients who move into the $D$ container due to ICU overload, together with the patients who move from $W$ to $D$. We write $D^{\text{Excess}}$ for the total number of these deaths.

Note that we have assumed that everyone who becomes infectious is eventually quarantined. While this assumption is probably unrealistically optimistic, it nevertheless means that our optimal shutdown strategy will provide a lower bound. We make similarly conservative choices with regard to the parameters contributing to the infection fatality ratio.

The disease part of the model involves a substantial number of parameters. Some of these are fairly well established but many remain quite uncertain. Table 1 summarizes the values and sources.

With these parameter values, the total infection fatality rate (IFR) without overload is

$$p_{\text{IFR}} = p_H \cdot p_{HD} + p_U \cdot p_{UD} = 0.48\%,$$

which is at the low end of the range of values reported in the literature [10].

**Table 1.** Parameter values and sources for the disease dynamics portion of the model, equations (2.1) and (2.2).

| parameter | value | remarks |
|---|---|---|
| $\alpha$ | 0.17 | Anderson *et al.* [9] |
| $\gamma$ | 0.2 | Anderson *et al.* [9] |
| $\mathcal{R}_0$ | 3.3 | Salje *et al.* [8] |
| $\beta$ | $\alpha\mathcal{R}_0$ | derived |
| $\kappa_M^{-1}$ | 14 | estimated[a] |
| $\kappa_H^{-1}$ | 10 | Salje *et al.* [8] |
| $\kappa_U^{-1}$ | 15 | Salje *et al.* [8] |
| $\kappa_W^{-1}$ | 2 | estimated[a] |
| $p_H$ | 0.021 | Salje *et al.* [8] |
| $p_U$ | 0.0047 | Salje *et al.* [8] |
| $p_M$ | $1 - p_H - p_U$ | derived |
| $p_{HD}$ | 0.17 | PHAC Emerging Sciences [10] |
| $p_{UD}$ | 0.27 | PHAC Emerging Sciences [10] |
| $p_{WD}$ | 0.8 | estimated[b] |
| $e_0$ | 200 | initial values of $I_0$ and $E_0$ |

[a]These estimated values are in line with the rates reported in Salje *et al.* [8].
[b]We were unable to find data for $p_{WD}$, and so we assumed that survival is very low for individuals needing a hospital bed but unable to access that resource.

## 2.2. Economic costs

We now describe the economic/control part of the model. To avoid complications, and the need to estimate a large number of parameters, we consider a one-sector economy, and use a nonlinear function to model the effects of different types of economic activity on transmission, following Chao [11]. See Chan *et al.* [12] for a much more detailed analysis of the effects of different kinds of non-pharmaceutical interventions on the control of COVID-19. The proportion of the economy that is shut down (SD) on day $t$ is given by $X_t \in [0, x_m]$. We take $x_m = 0.5$. Here $x_m < 1$, as parts of the economy, e.g. food distribution, cannot be shut down. We assume that the shutdown is arranged so that the parts of the economy with the biggest effect on the transmission rate $\beta$ are chosen first, and thus we obtain a concave effectiveness from the shutdown.

Let $f(x)$ be the proportional reduction in social contacts due to the shutdown; we call $f$ the *shutdown effectiveness* function. A shutdown of $X_t$ will lead to an infection rate on day $t$ of

$$\beta_t = \beta(1 - f(X_t)).$$

We do not have the data to calculate $f$ in detail. A simple model is to take

$$f(x) = x^\theta, \tag{2.3}$$

(figure 2) where $\theta < 1$.

The reproduction number due to a SD of $x$ is then given by

$$\mathcal{R}(x) = \mathcal{R}_0(1 - x^\theta). \tag{2.4}$$

We note that $d\mathcal{R}_0(0+)/dx = -\infty$, which implies that some level of shutdown is always advantageous—see appendix B.

With an effectiveness function of this form we need to estimate $\theta$. A first guess would be to use the 'Pareto principle', which states roughly that '80% of the output is due to 20% of the input'. We thus write $f(1/5) = 4/5$ which gives $\theta \approx 0.14$. This guess has some support from data: the fall in GDP in Canada in the first quarter of 2020 is estimated to lie between 7% and 14% [13–15]. Anderson *et al.* [9, p. 10] estimates that the March–April 2020 shutdown in BC reduced contacts by about 78%, with a 90% credible interval of 66% to 89%, which reduced $\mathcal{R}(x)$ to below 1. Using these values for GDP and

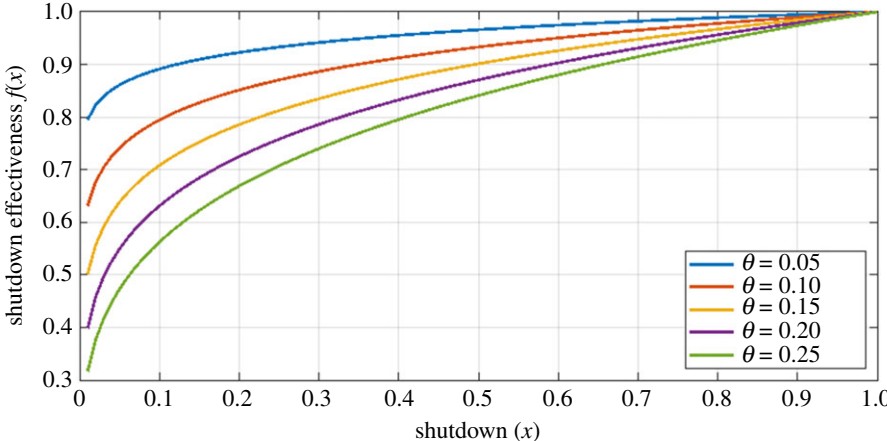

**Figure 2.** The shutdown effectiveness function, $f(x) = x^\theta$. The higher the value of $\theta$, the greater is the reduction in effective reproduction number for a given level $x$ of economic shutdown.

contact reduction in (2.3), we obtain $\theta \in [0.04, 0.21]$. The highest values of $\theta$ are obtained for the highest fall in GDP, and the lowest reduction in contacts. We note that the GDP estimates do not include capital destruction costs, which tend to take longer to become evident [16], and so we assume that the ultimate cost is actually higher than estimated. It therefore seems prudent to set $\theta$ at a value closer to the upper end of the plausible range. We thus select $\theta = 0.2$ as the default value.

We write $x_c$ for the amount of SD needed to make $\mathcal{R}$ equal to 1; thus

$$x_c = \left(\frac{\mathcal{R}_0 - 1}{\mathcal{R}_0}\right)^{1/\theta}. \tag{2.5}$$

If $\mathcal{R}_0 = 3.3$ and $\theta = 0.2$, then $x_c = 0.164$. Given the initial conditions, the control $X_t$ and the vaccinations $v_t$, the equations above describe the evolution of the epidemic for $t \geq 0$. Once we have introduced costs, as described below, we can then search for a strategy $(X_t)$ which will minimize costs.

A standard difficulty with optimization of a temporal process is the choice of a terminal time. Suppose first that we consider the model as given above over a fixed time window $0 \leq t \leq T$. For many parameter values, the optimum strategy is to control the epidemic quite strictly until shortly before time $T$, and then to relax. This leads to a rapid growth of infections just before the terminal time $T$; the costs of these infections (death and medical) are not counted as they occur after time $T$. Clearly, however, these post-$T$ infection and economic costs should be included. We chose to handle this difficulty by introducing a very simplified model of vaccinations, which we assumed would ultimately cover the whole population, and be 100% effective.

We take $T_V$ to be 360 days; this is the approximate time from the start of the epidemic (March 2020) until we might hope that a vaccine is available. We assume that the vaccine is 100% effective and that $N_V$ people are vaccinated daily for each day $t \geq T_V$ under the terminal time $T = T_V + N/N_V$. By time $T$ the whole population is vaccinated and the epidemic is at an end. Thus we set $v_t = 0$ for $1 \leq t \leq T_V$, and $v_t = \min(N_V, S_t(1 - \beta_t I_t/N))$ for $T_V + 1 \leq t \leq T$.

Herd immunity will be reached before the terminal time $T$ [17], but even after herd immunity is achieved infections, and hence costs, will still occur. It may be advantageous to impose some measure of shutdown after herd immunity is reached—this shutdown leading to a quicker end to the whole epidemic. With our base parameter values we have $T_V = 360$ and $T = 610$.

We now introduce costs; throughout the paper these are in Canadian dollars (CAD). The costs we consider are the economic costs of the shutdown $C^{(SD)}$, costs arising from the medical care of infected individuals $C^{(M)}$, and costs due to lives lost $C^{(D)}$. All these costs are counted over the whole period of the model, up to the time $T$.

The economic cost of shutdown is taken to be the daily shutdown costs summed over the time period of the epidemic, so that writing $G_D$ for the daily GDP of BC we have

$$C^{(SD)} = \sum_{t=1}^{T} G_D X_t.$$

**Table 2.** Economic parameters. Dollar values are in Canadian dollars. For the lost wages per day of illness, $v_M$, we take the drop in GDP and divide by the number of days over which the drop occurred. For the number of daily vaccinations after $T_V$, we take the population of BC [22] divided by the estimated length of time it will take to vaccinate everyone in the province [23].

| parameter | value | remarks |
| --- | --- | --- |
| $N$ | 5 100 000 | population of BC [22] |
| $G_D$ | $800 000 000 | daily GDP in BC [15] |
| $v_M$ | $100 | lost wages per day of illness |
| $v_H$ | $1000 | day in hospital [18] |
| $v_U$ | $2000 | day in ICU [18] |
| $v_{NU}$ | $8000 | ICU set-up [18] |
| $V_L$ | $7 000 000 | value of life [24] |
| $T_V$ | 360 | number of days until vaccination starts |
| $N_V$ | 20 400 | number of daily vaccinations after $T_V$ |
| $T$ | $T_V + N/N_V$ | terminal time for model |
| $\theta$ | 0.2 | effectiveness of shutdown: range 0.1–0.2. |
| $x_m$ | 0.5 | maximum shutdown allowed |

Medical costs can be inferred from Jones [18]; we take the cost of a day in a regular hospital bed as $1000, and ICU costs as $10 000 on the first day and $2000 per day thereafter. The medical cost during the period $t = 0, \ldots, T$ is therefore

$$C^{(M)} = \sum_{t=1}^{T} (v_M M_t + v_H H_t + v_U U_t + v_{NU} U_t^{New});$$

here $v_M$, $v_H$, $v_U$ are the daily costs for patients in states $M$, $H$ or $U$, $v_{NU}$ is the excess cost for the first day in ICU, and $U_t^{New}$ is the number of new patients in ICU on day $t$.

To determine costs due to the lives lost, we need to determine the 'statistical (dollar) value of a life' (SVL) or $V_L$. There is a substantial literature on this topic—see Dionne & Lanoie [19], Greenstone & Nigam [20], and Thunström *et al.* [21]; authors use a variety of methods to get some handle on this number. As one would expect, estimates vary widely—the review in Dionne & Lanoie [19] gives a range for Canada of C$2.0m—C$11.1m, with a median of C$5.5m. In 2020 dollars that gives $V_L =$ C$7.0m, which we take as our base figure. We then set

$$C^{(D)} = V_L D_T.$$

The total cost of the epidemic is the sum of the shutdown, medical and death costs:

$$C^{(Tot)} = C^{(SD)} + C^{(M)} + C^{(D)}. \tag{2.6}$$

Note that the first two of the costs in (2.6) are pure dollar costs, while the third is based on a more subjective evaluation of the dollar cost of a lost life. We use the SVL in order to estimate the total cost of the epidemic, but since the quantitative assessment of the pure dollar and SVL costs are so different, in part of our analysis below we also treat these two types of costs separately. We write

$$C^{(Dol)} = C^{(SD)} + C^{(M)}. \tag{2.7}$$

It will be seen from the above that we consider only immediate or short-term costs of the epidemic; our model implicitly assumes that once the epidemic is over and the shutdown is reduced to zero, the economy springs back to its pre-pandemic state. No doubt there will actually be many long-term costs, and very likely also some long-term gains, from the pandemic. An example of the former is capital destruction due to business closures, and the latter increases in efficiency due to the introduction of new business methods. At this point, these costs are extremely uncertain, and we have not attempted to include them in our model. We discuss these additional costs further in §4.3.

We list the economic parameters in table 2.

## 2.3. Optimization procedure

We considered two types of optimization. In §3.1, we minimized the total cost of the epidemic using value of life $V_L$ to translate lost lives into costs. Since it does not always make sense to translate a lost life into a dollar value, we take a different approach in §3.3. There we looked at Pareto optimization [25] of pure dollar costs and deaths, i.e. for the two numbers $(C^{(\mathrm{Dol})}, D_T)$. We studied the structure of the epidemic associated with some sample points on the Pareto curve.

In both cases, we only considered strategies in which $X_t$ was held constant for a significant time period. For the total cost optimization we divided the time period into 10 equal segments of 61 days (about two months), and held $X_t$ constant in each of these periods. Writing $x_i$ for the value of $X_t$ in period $i$, the strategy $(X_t)$ is described by the vector $x = (x_1, \ldots, x_{10})$. We used Matlab's multivariate optimization routine 'fmincon' to optimize over vectors $x$. For the Pareto optimization we used five equal time periods, each of 122 days (the Pareto optimization performed poorly with 10 periods).

We had several reasons for this restriction on shutdown strategies. First, from a computational point of view, it is not feasible to optimize over several hundred control values $X_i$. Next, since the effectiveness function $f$ is concave, rapidly varying control strategies will perform less well than more slowly varying control strategies (see appendix A). Finally, it is not realistic to consider that a government could implement a shutdown function $X_t$ that changes too frequently.

We remark that it would be as computationally feasible to consider piecewise linear strategies as piecewise constant ones—in both cases the optimization reduces to optimizing over a 10-dimensional space (if we use our baseline set-up). However, we did not consider such strategies because we did not consider them to be applicable in practice.

The two types of optimization are connected, since the total cost optimization corresponds to tangents on the Pareto curve. More precisely, write $(C, D)$ for the two axes (dollar costs and deaths) for the Pareto curve. Let $vD + C = b$ be a tangent to the Pareto curve at the point $(C_1, D_1)$. Then $b = C_1 + vD_1$ is also the minimum total cost when we take $V_L = v$.

# 3. Results

Before we consider the optimization problem in detail, it is helpful to look at the basic structure of the model described above. If the epidemic is such that there is no overload of hospital capacity, and so no excess deaths, then the total number of infected individuals is $R_T + D_T$, and the total number of deaths is

$$D_T = p_{\mathrm{IFR}}(R_T + D_T).$$

We can compute the expected 'value of life' costs as the probability of dying times the value of a lost life, i.e. $V_L p_{\mathrm{IFR}} = \$33\,600$, for our baseline parameter values. On the other hand, the expected medical costs are

$$p_M \kappa_M^{-1} v_M + p_H \kappa_H^{-1} v_H + p_U (v_{NU} + \kappa_U^{-1} v_U) = \$1574,$$

for the baseline parameter values. Thus for most parameter values the medical costs of the epidemic are small in comparison with the value of life costs.

## 3.1. Optimization of total costs

### 3.1.1. Baseline (constant shutdown) scenarios

The simplest strategy is to take $X_t = x$ constant for the whole period. Figure 3 shows the total cost as a function of $x$ for various values of the SVL $V_L$. This graph has several informative features. We consider first the curves with $V_L \geq \$3.0$m. In these cases, the minimum value $x_{\min}$ is a little smaller than $x_c = 0.164$ (if $V_L = \$7.0$m then $x_{\min} = 0.13$). These values of $x$ correspond to allowing the epidemic to grow slowly over the pre-vaccination period; it then declines once vaccination reduces the effective reproduction rate. One sees further that the minimizing value $x_{\min}$ is not very sensitive to $V_L$. These values of $x_{\min}$ give rise to a relatively small epidemic, so any further increase in $x$ would save relatively few lives. For $x > x_c$, the total cost curves for differing values of $V_L$ are very close together; this is because in this regime the total number of deaths is small (a few hundred), so the term $V_L D_T$ is much smaller than the pure dollar cost $C^{\mathrm{Dol}}$.

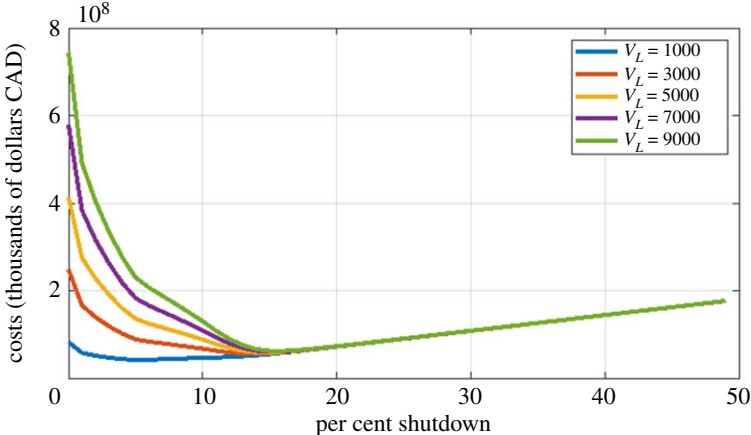

**Figure 3.** Total costs as a function of shutdown level for a constant shutdown strategy. Several values of $V_L$ (in thousands of dollars CAD) are shown.

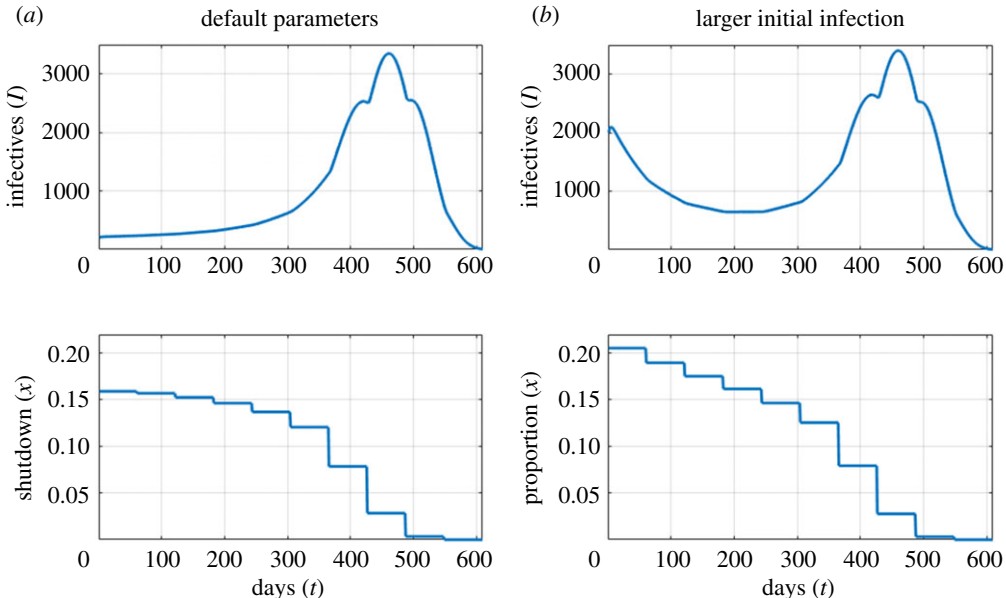

**Figure 4.** Number of infections and shutdown for optimal policy for base parameters ($a$) and a larger initial epidemic with $I_0 = E_0 = 2000$ ($b$), minimizing total costs, $C^{(Tot)}$ (see (2.6)). For these parameter values, the critical shutdown level is $X_c = 0.164$. Vaccination begins at $T_V = 360$.

For $V_L = 1.0$m, one sees a different pattern. The minimum cost is obtained at a much lower shutdown level, around 5%. In this case, deaths are counted as relatively unimportant, so the optimal strategy is to adopt a relatively mild shutdown.

### 3.1.2. Variable shutdown scenarios

We next consider strategies that are piecewise constant, being allowed to vary only every 61 days. We see the same general pattern as with a constant shutdown. For the baseline parameter values, we find that the optimum policy (figure 4$a$) starts with a fairly strict shutdown, with $X_t \approx x_c = 0.164$, for the first six time periods. Subsequently, $X_t$ is gradually relaxed, and is zero in the final period. Taking $X_t < x_c$ means that the epidemic grows over the first year, but at a slow and controlled rate. As vaccinations proceed, the optimum policy is to relax the shutdown enough to allow a late peak in the epidemic. This peak occurs during the period when the vaccine has become available. If the initial infection level is 10 times higher than the baseline (figure 4$b$), we see a similar pattern, but with higher shutdown levels at the beginning, and shutdown levels dropping more in the first third of the year.

**Table 3.** Economic and epidemiological outcomes corresponding to the optimal solution of the model under variation of a few key parameters (left column). The total cost outcome includes the value of lives lost (all dollar values are in CAD). For each row in the table, all parameters except the one being varied are held at the default values listed in tables 1 and 2. For comparison, the outcomes under default values for all of the parameters are listed in the first row.

| parameter(s) varied | total cost ($ billion) | deaths | max infected | excess deaths |
|---|---|---|---|---|
| None | 51.4 | 494 | 3345 | 0 |
| $V_L = \$5.0m$ | 50.3 | 697 | 4783 | 0 |
| $V_L = \$3.0m$ | 48.4 | 1228 | 8435 | 0 |
| $V_L = \$2.0m$ | 46.9 | 2214 | 13 835 | 0 |
| $V_L = \$1.0m$ | 34.1 | 20 477 | 161 167 | 8751 |
| $V_L = \$0.0m$ | 6.8 | 64 339 | 584 296 | 55 932 |
| $\theta = 0.15$ | 27.8 | 347 | 2124 | 0 |
| $\mathcal{R}_0 = 3.0$ | 40.5 | 445 | 2936 | 0 |
| $\mathcal{R}_0 = 3.6$ | 62.5 | 532 | 3657 | 0 |
| $\mathcal{R}_0 = 5.0$ | 112.1 | 576 | 4591 | 0 |
| $\mathcal{R}_0 = 8.0$ | 191.9 | 4518 | 25 891 | 0 |
| $\mathcal{R}_0 = 8.0, V_L = \$3.0m$ | 143.4 | 29 419 | 163 447 | 13 522 |
| $e_0 = 2000$ | 59.0 | 665 | 3,408 | 0 |
| $e_0 = 20\,000$ | 68.6 | 1250 | 20 782 | 0 |
| $N_V = 40\,800$ | 46.27 | 472 | 5117 | 0 |
| 20 time periods | 51.3 | 494 | 3105 | 0 |
| 1 time period | 69.6 | 811 | 5341 | 0 |

Note that in this second scenario, the optimal strategy designates $X_t > x_c$ initially, so that the epidemic decreases, bringing the case load under control before resuming the late wave pattern of the baseline case.

The lower plots in figure 4 show the shutdown strategies, which are piecewise constant, with jump discontinuities. A change in shutdown leads almost immediately to a change in the infection rate, which explains why one sees some discontinuities in the derivative of the infections. These discontinuities will always be present, but are only visible for large changes in the shutdown level.

In all the cases we examined, the optimum shutdown was unimodal. Further, for parameter values close to our baseline ones the maximum shutdown was at the start of the epidemic, with $X_t$ decreasing in small steps at first, and then in larger steps as a vaccine becomes available. Alternating periods of high and low levels of shutdown did not occur. See table 3 for a summary of the outcomes, in terms of deaths, epidemic size and costs. Increasing the number of time periods to 20 makes little difference to the total cost or number of deaths. When $V_L \geq \$2.0m$ the shutdown is severe enough to control the epidemic and avoid overloading the medical system; thus in these cases the number of excess deaths is zero.

As in the constant policy case, for $V_L \leq \$1.0m$ the optimal strategy changes to one which permits a large epidemic, with many deaths. The shutdown values for the first three periods are 0.0001, 0.0684, 0.0718. One might expect that a strategy which is the same for the final seven periods, but is the average of the first three at the beginning (i.e. 0.0467), would perform better, but this is not the case. This modification to the optimum strategy delays the epidemic peak, but gives less social distancing when it is needed most, and allows a bigger epidemic, with 31 010 deaths as opposed to 20 477 (figure 5).

Thus, apart from the need to control a large initial epidemic, one finds one wants $X_t$ close to $x_c$ in the pre-vaccination phase. Taking $X_t$ significantly smaller than $x_c$ causes a large epidemic, with many deaths, while taking $X_t$ significantly larger than $x_c$ is economically costly, and saves few lives.

## 3.2. Sensitivity analysis

Since some of the parameters are not well known, we tested our results using a partial rank correlation coefficient (PRCC) sensitivity analysis as described by Marino *et al.* [26] or Pianosi *et al.* [27]. PRCC is a reliable measure of the contribution that parameters have to the model when the relationship between the

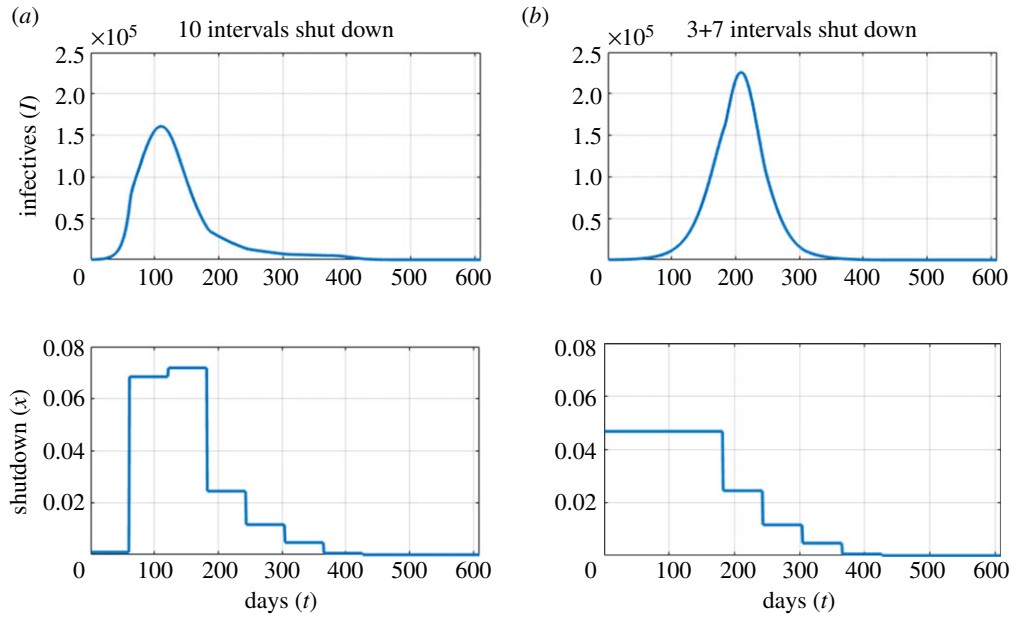

**Figure 5.** Effect of a lower $V_L = \$1.0$m. Number of infections and shutdown for the optimal policy (*a*), and a modified control held constant at the average value of the optimal control for the first three time periods (*b*). The number of deaths for the optimal control is 20 477, and for the modified control is 31 010.

parameter and the output is monotonic. Each parameter is varied over a range from half to twice the baseline value given in tables 1 and 2. The PRCC values can be interpreted as the correlation between parameter and model output, linearly discounting the effects of the other parameters.

We ran the sensitivity analysis on the model with the optimal strategy given for the baseline parameters, and considered the sensitivity (i) with respect to both total costs (i.e. the sum of pure dollar and value of life costs), and (ii) with respect to deaths. Both sensitivity analyses show high (greater than 0.6) sensitivity values for the parameters $\mathcal{R}_0$ and $\theta$. In addition, for costs $T_V$ and $N_V$ had moderate sensitivity, in the range (0.2–0.6). For deaths, the parameters $p_H$, $p_{HD}$, $p_{UD}$ and $e_0$ had moderate sensitivity. See the electronic supplementary material for full details. In both cases, the parameters $\alpha$, $\gamma$, $\kappa_M^{-1}$, $\kappa_H^{-1}$, $\kappa_U^{-1}$, $\kappa_W^{-1}$, $p_{UD}$, $p_{WD}$ had low sensitivity—less than 0.11.

It may seem somewhat surprising that $V_L$ has a relatively low sensitivity of 0.19. One reason for this is that our sensitivity analysis only studies sensitivity of a parameter $x$ over the range $[x_0/2, 2x_0]$, where $x_0$ is our base value. Thus for $V_L$, the sensitivity analysis only considered the range between \$3.5m and \$14m, and, as we already saw when we looked at constant shutdowns, in this range the value of life makes little difference to the strategy or to the total cost.

As well as the PRCC sensitivity analysis described above, we wished to understand the effect of some key parameters on the overall form of the epidemic—that is, their effect on the optimal shutdown, costs, deaths and excess deaths. We therefore ran a number of additional scenarios, and table 3 lists the economic and epidemiological outcomes for some of these. We do not see excess deaths (and a large decrease in cost) until $V_L$ drops to \$1.0m or less. Increasing the frequency with which the control strategy is adjusted (i.e. increasing the number of time periods) has very little effect on any of the outcomes. By contrast, decreasing the adjustment frequency down to the point where only one constant shutdown level is used throughout the epidemic results in almost twice as many deaths and increases the total cost. Decreasing the parameter $\theta$ and basic reproduction number $\mathcal{R}_0$ both lead to reduced cost and deaths, while increasing $\mathcal{R}_0$ and the initial disease prevalence $e_0$ both lead to increased costs and deaths. Doubling the vaccination rate, and hence reducing the total time period $T$, had little effect on the basic strategy, but, not surprisingly, led to lower costs and deaths.

We considered some scenarios with high $\mathcal{R}_0$. If $\mathcal{R}_0 = 8.0$ then $x_c = 0.51$, which is greater than the maximum allowed shutdown $x_m = 0.5$. In this case the optimal strategy is to take $X_t = 0.5$ for nearly the whole period up to the time $T_V$ when vaccinations begin. It is interesting that even in this case, with a very large $R_0$, the optimal strategy is still to control infections strictly. If, however, we also reduce $V_L$ to \$3.0m then the optimal control 'flips' to one which allows a large epidemic.

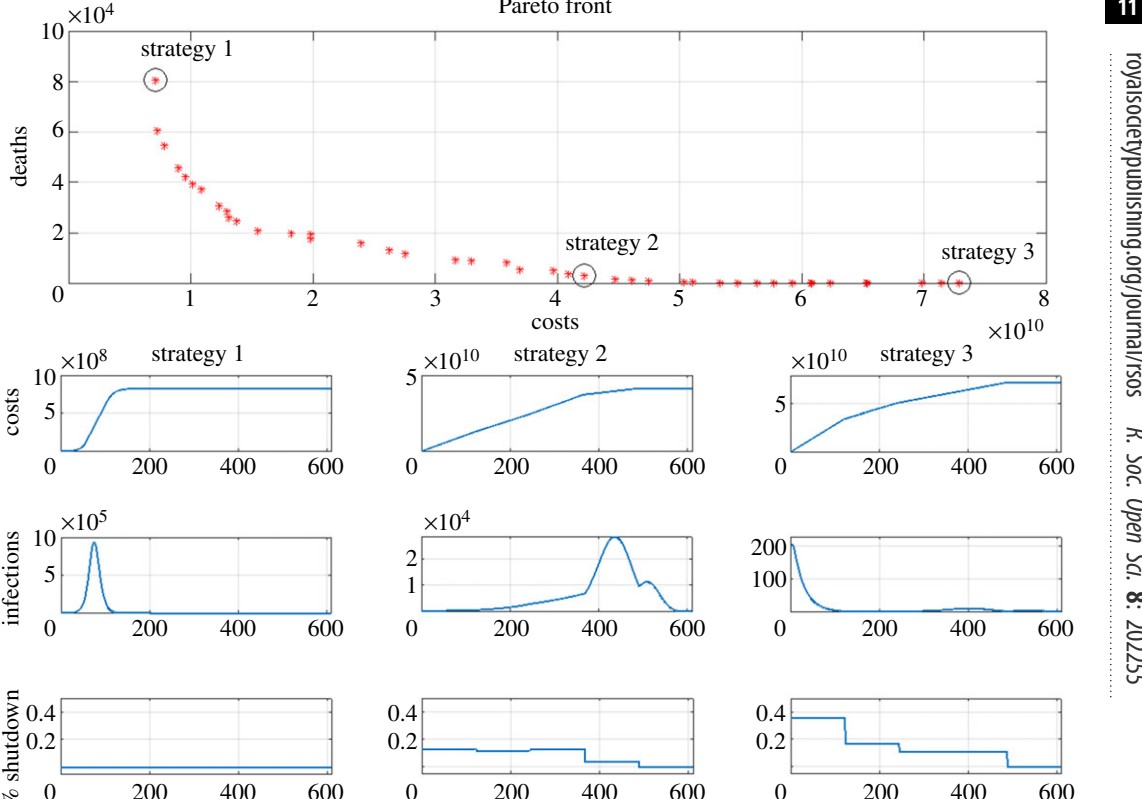

**Figure 6.** Top: Pareto plot. Each asterisk corresponds to the shutdown strategy yielding the approximately minimum possible number of deaths $D_t$ at dollar cost $C^{(Dol)}$ (i.e. not including the value of a lost life). The Pareto front assumes default parameter values and five shutdown intervals. Strategy 1: minimize costs only. Strategy 2: moderate shutdown. Strategy 3: minimize deaths only. Bottom grid of plots: cumulative cost (top row), daily infections (middle row), and daily shutdown strategy (bottom row).

## 3.3. Pareto optimization: costs and deaths

In the results presented above, we find the optimum shutdown pattern by expressing all of the constraints, including lives lost, in terms of their monetary value, $C^{(Tot)}$. All costs are set to the default values given in table 2. Here, we separate out pure dollar costs $C^{(Dol)}$ from deaths, applying the two as separate constraints, and present the Pareto optimum [25]. The results are shown in figure 6. The curve shows the competing effects of minimizing costs versus minimizing deaths: Minimizing the pure dollar costs of shutdown leads to higher mortality, and minimizing mortality leads to higher pure dollar costs. We remark that the Matlab Pareto optimization routine performed much less well than the simple optimization, and so one should not rely on the finer details of figure 6. The overall shape, however, is correct.

Each point $(x, y)$ on the Pareto curve corresponds approximately to the shutdown strategy which gives the minimum possible number of deaths $y$ at pure dollar cost $x$. We have highlighted three of the strategies on Pareto-optimum curve; the shutdown strategies are shown in the lower half of figure 6, and the Pareto points they correspond to are circled on the Pareto curve. Strategy 1 takes $X_t$ very small and has a dollar cost of $7.13 billion, about 80 000 deaths, of which about 70 000 are 'excess deaths' due to overload of the medical system. We note that the Pareto optimization fails to find a slightly better strategy, shown in table 3 in the line corresponding to $V_L = 0$ (this strategy takes $X_t = 0.0009$ for the first time period). Strategy 3 prioritizes minimizing deaths over costs. In this case, the shutdown level is initially very high—about 40%, and is then decreased at each subsequent time period until it becomes zero in the last interval. The dollar cost skyrockets to over $70 billion, and the number of deaths is less than 100. Strategy 2 is an intermediate strategy. If we choose this strategy, pure dollar costs can be reduced to near half that of the most expensive strategy, and the total deaths are a few thousand.

We note here that as $\mathcal{R}_t$ increases, the Pareto front straightens out, becoming a very nearly straight line joining Strategies 1 and 2 for $\mathcal{R}_t = 7.t$, which is the estimated value for the Delta variant [28]. At this level

of infectiousness, governments have no obvious optimum available, where costs are not astronomical and deaths are still fairly low. Pareto fronts for higher values of $\mathcal{R}_I$ can be found in the appendix.

# 4. Discussion

Our study is motivated by the hesitation shown by some leaders to implementing strict control measures to slow the spread of COVID-19 [29–31], in part due to the huge toll to the economy. For informed decision making, it is clear that we need some objective quantification of the total cost of both the health crisis and the economic shutdown measures. In this work, we present a coupled disease and economic cost model which is a useful tool for evaluating shutdown options. While our model is in no way a comprehensive representation of all of the costs and benefits of shutdown measures, we submit that it contains the salient features of the system, and so the patterns in our results should reflect real dynamics.

For a simple minimization, it is necessary to combine our two key variables, that is the dollar cost (medical costs plus economic shutdown costs) and deaths, which we did by using the 'value of a statistical life' $V_L$. For values of $V_L$ close to the consensus value of about \$7.0m given in Dionne & Lanoie [19] our results indicate that total costs (deaths, hospital costs and deaths costs) are minimized by a significant shutdown—of about 10–15% in the early stages of the epidemic. For these values of $V_L$, the optimal shutdown strategy decreases slowly in the initial phases of the epidemic, and then falls rapidly to nearly zero once vaccination is well underway and the system is close to herd immunity. In the middle of the vaccination phase, the shutdown is relaxed sufficiently to allow a moderate late peak in infections. At this point, the effective reproduction number has been reduced sufficiently so that the epidemic has no possibility to explode before the completion of the vaccination programme.

If $V_L$ is taken to be \$1m or less, a different pattern occurs. As death costs are smaller, economic costs play a relatively larger role, and the question is how best to deploy the relatively small amount of shutdown that will be introduced. It turns out that the optimal control is to wait until the infections are rising rapidly, and then use the shutdown to reduce the height of the epidemic peak.

In this paper, we have studied the optimization problem for the situation when the disease and economic parameters are known. In practice of course this is not the case, and in particular neither of the two most significant parameters ($\mathcal{R}_0$ and $\theta$) would have been known to a government considering lockdowns. One therefore needs to consider strategies using feedback to control the epidemic. In the context of our model, a reasonable goal would be to aim at having the effective reproduction number $\mathcal{R}_0(X_t) = \mathcal{R}_0(1 - X_t^\theta)$ close to 1. The results of Stewart et al. [32] suggest this goal is reasonably attainable.

## 4.1. Gradual versus periodic shutdowns

Our observations provide an interesting perspective on the shutdown approaches taken by governments in BC and elsewhere. In almost all cases, after a severe initial shutdown, economies were reopened and it was hoped the transmission rate could be controlled through contact tracing as well as individual prophylactic behaviours (mask-wearing, washing hands, avoiding crowds, keeping business patrons 2 m (6′) apart, etc.). Unfortunately, these methods do not seem to be as effective as was hoped, and as we write (November 2020) cases are rising dramatically in many jurisdictions, and governments are reimplementing significant shutdown measures [1]. From our work, it appears that a slower and more gradual decrease in shutdown level would have led to a smaller overall cost of the epidemic.

## 4.2. Cost of lives lost

Since the equating of a life lost to a dollar value is debatable, we found it useful to separate pure dollar costs and deaths. The Pareto curve shows us that deaths can be maintained at a very low level, even if the shutdown level falls short of 'stop the epidemic'. Again, unless it is acceptable to let the number of deaths skyrocket, the optimal shutdown strategy always includes a positive and significant (i.e. over approximately 15%) level of shutdown. Wulkow et al. [33] obtained a similar result in a model where 'shutdown' was measured with respect to mask-wearing, school closures and contact tracing.

In this work, we use a single value for the value of life $V_L$: that is, we do not make any adjustments for the age of the deceased. If we neglect excess deaths, then this gives a SVL cost of $C_0 = p_{IFR} V_L$ per infection; with our base parameters this is \$36 400. It is natural to ask what happens if we look at an age-stratified

population. Mortality rates are substantially higher for older patients [8], and as they have a smaller life expectancy it can be argued that their SVL should be smaller.

To assess this point, we can look at the following simple age-stratified model. We divide the population into $n$ age groups. Group $i$ consists of a proportion $a_i$ of the population, has death rate (IFR) $p_i$ and has SVL $v_i$. Then if we neglect excess deaths, the overall cost per infection is

$$C_1 = \sum_{i=1}^{n} a_i p_i v_i.$$

Let $v_i$ for each age group be the life expectancy of the mid point of the age range times \$170 000, which we take to be the value of a year of life. Using the age-related death rates from Salje *et al.* [8] and BC census data from 2011, we obtain $C_1 = \$12\,377$, i.e. about 38% of $C_0 = p_{\mathrm{IFR}} V_L = \$33\,600$. Thus the age-stratified model outlined above corresponds to taking $V_L$ to be 37% of the value listed in table 2, i.e. about \$2 600 000. (See the electronic supplementary material for more details.) The outcomes listed in table 3 show that for $V_L$ as low as \$2.0m, the optimal shutdown strategy is sufficiently large to prevent any excess deaths, and so, broadly speaking, our results are not changed. Note that this simple calculation neglects excess deaths and assumes that all the disease parameters except death rates are independent of age.

A more complete treatment of the issue of age would require an age-segregated model, which would be significantly more complicated than the one given here, and would need many more parameters.

## 4.3. Other costs

Since our focus is the epidemic in BC, we make a few remarks about the specific situation there, most notably, the overdose crisis. While our work strongly suggests that the adoption of economic shutdown measures in BC is, in the long term, a responsible strategy, the shutdown in BC has caused a significant increase in deaths due to drug overdoses there [34]; the excess (total overdose deaths over and above the usual average) for March to June 2020, exceeds the total number of COVID-19 deaths for that time period by over 40% [34,35]. A number of factors are probably involved in this mortality, including the disruption of regular drug supply chains, through border closures, leading to an increasingly toxic drug supply [36], the reduction in access to harm reduction services as a result of physical distancing protocols [36], and increased stress resulting from increased isolation and economic uncertainty [37–39] (similar patterns have been observed in the USA [40]). The value of these lost lives could have a significant effect on our cost calculations, but assessing the relation between the level of economic shutdown and the number of overdose deaths is not straightforward. The social issues around the legality of drugs, addictions treatment and overdose deaths is well beyond the scope of this paper, but is nevertheless an extremely important issue.

There are many additional costs to social distancing that have not been included in our model due to lack of data. We discuss some of the most important of these here. First, while our economic cost calculations take account of loss of wages and business income due to the shutdown, it does not include costs due to capital destruction as there continues to be considerable uncertainty in this area. The Canadian Federation of Independent Business estimated in July 2020 that 12% of small and medium-sized BC businesses were at risk of closure [41,42]; a year later, 79% of BC small businesses were fully open with approximately half seeing normal sales [43]. How many businesses will be able to open, or remain open, after the pandemic ends is still unclear, and so it is difficult to estimate the effect of these possible closures on provincial GDP.

Second, with hospitals stretched to accommodate COVID-19 patients, and with fear of contracting COVID-19 limiting movement, treatment of non-COVID illnesses can be severely delayed [44,45]. Thus there will be excess deaths from other causes, and costs associated with suffering due to untreated conditions [44,46]. Finally, mental health, in general, is known to suffer during community disasters [47,48], including pandemics. Research is emerging regarding the effects of COVID-19 non-pharmaceutical measures on mental health [37–39,49,50], but costs are difficult to determine at this stage.

## 4.4. Future considerations

Our understanding of the SARS-CoV-2 virus and COVID-19 disease are continually evolving, and so necessarily our model is relevant to the current epidemic up to a certain time point. In particular, we do not consider the possibility of a successful treatment that reduces mortality among those who

become infected. There is evidence that treatment outcomes are improving for COVID-19 patients [51], but the data are still preliminary and therefore difficult to include. We also do not include contact tracing as a measure for reducing transmission. Considerable work evaluating the effectiveness of contact tracing has been done by other researchers [52,53], and the general consensus is that contact tracing alone is insufficient and can quickly become overwhelmed by a few large-scale outbreaks.

Finally, we remark that the eventual cost of the epidemic depends a great deal on the time horizon considered, and the value a society places on its elderly population (the group most at risk) [54]. Indeed, the metric used to evaluate a country's economy is another factor that could change the calculation significantly [55]. These considerations are beyond the scope of the current paper, but provide interesting avenues of future work in this area, and would help countries prepare for future pandemics.

We also remark that this paper considers the management of COVID-19 as a problem with a definite endpoint, which is taken to be when the epidemic vanishes from the population due to vaccination and herd immunity. In fact, it seems clear that the initial 'acute' phase of the epidemic will be followed by a long-term management problem as new variants arise and immunity due to vaccination decays. We have little information at present on the magnitude of these effects, and these longer-term questions are outside the scope of this paper.

# 5. Conclusion

Our analysis suggests that the BC and federal governments were wise to impose severe shutdown levels at the beginning of the epidemic. The later reduction in shutdown levels was probably too large when compared with an optimal strategy. While our model is relatively simple and entirely theoretical, our results are consistent with other, more complex models carefully validated against real-world data [12,33,56]. Our approach is therefore useful for developing intuition, and uncovers the fundamental trade-offs between controlling disease-related deaths and economic costs.

The difficulty with maintaining shutdown levels, as per the optimum strategy, is that people are unlikely to cooperate with rules that appear to be unnecessarily restrictive, especially when livelihoods are at stake [57,58]. Future modelling studies that include behavioural components will be very helpful. Much work needs to be done by government and society to make sure that not only are countries economically prepared to handle future pandemics, but that populations also have the necessary understanding and mental resiliency to maintain a high level of prophylaxis for a long period of time.

Data accessibility. The Matlab code used to study the model presented in this paper is publicly available on Zenodo at https://zenodo.org/record/5275461#.YSgMYlsTFH5.

Authors' contributions. M.T.B. developed the initial model, contributed to the coding and simulation study, wrote part of the first draft and edited the paper. N.D.M. developed most of the code, did most of the simulations, developed the figures, and wrote the first draft of the paper. R.C.T. contributed to the coding and simulation study, finalized figure design and edited the paper.

Competing interests. We declare we have no competing interests.

Funding. This work was supported by (R.C.T.) NSERC DG RGPIN-2016-05277 and (M.T.B.) NSERC DG RGPIN-2016-03703.

Acknowledgements. The authors would like to thank the BC Covid Research Group for helpful discussions, and two anonymous reviewers for their insights. This work was done on the traditional, ancestral and unceded territory of (R.C.T.) the Sylix (Okanagan) Nation, and (M.T.B.) the Musqueam peoples.

# Appendix A. Slowly varying shutdown is better when $f$ is concave

In §2.3, we stated that slowly varying SD strategies would outperform rapidly varying strategies because the shutdown effectiveness function $f$ is concave. To see this relationship, consider a strategy with period $2S$, where for the first half of each period $X_t \equiv x_1$, and $X_t \equiv x_2$ for the second half of each period. If the $x_i$ are close enough to $x_c$ so that the epidemic does not explode over a period of $S$ days, then an analysis of the SEIQ equation shows that the effect of the shutdown is roughly linear, so that the periodic strategy above has approximately the same effect as a constant strategy $x'$ with $f(x') = \frac{1}{2}(f(x_1) + f(x_2))$. As $f$ is

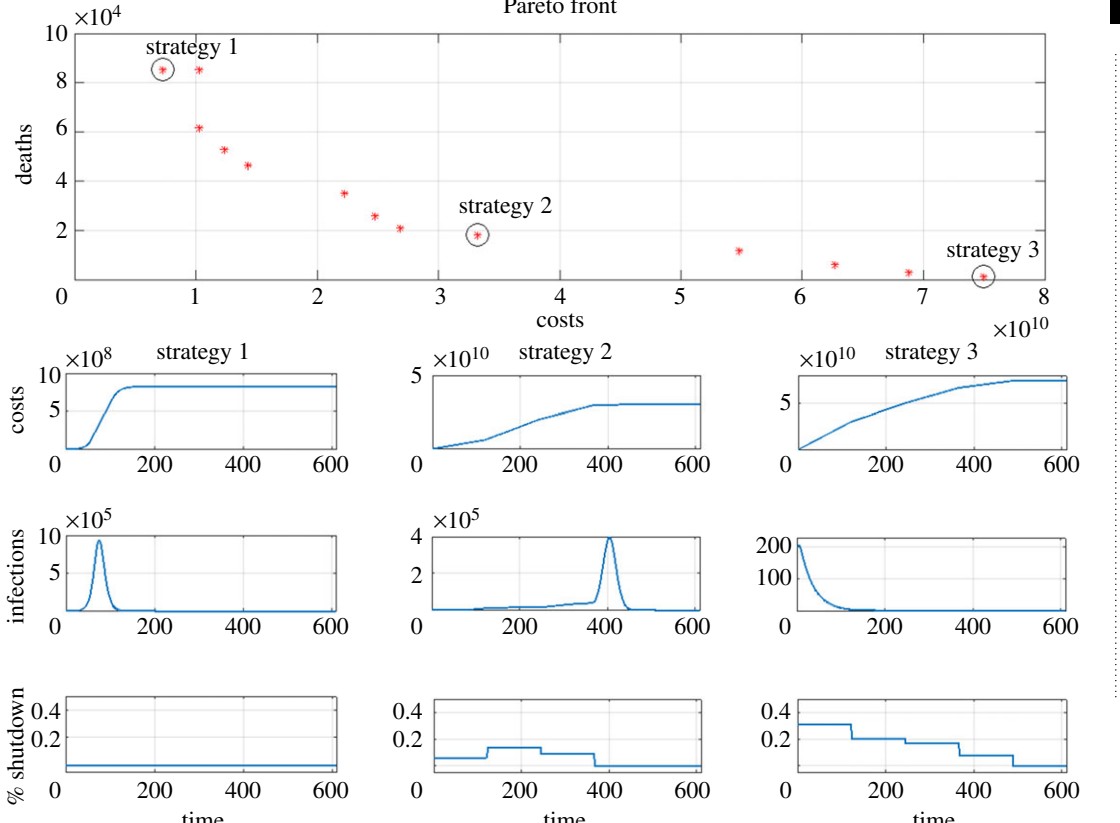

**Figure 7.** Top: Pareto plot for $\mathcal{R}_0 = 4.0$. Each asterisk corresponds to the shutdown strategy yielding the approximately minimum possible number of deaths $D_t$ at dollar cost $C^{(\text{Dol})}$ (i.e. not including the value of a lost life). The Pareto front assumes default parameter values and five shutdown intervals. Strategy 1: minimize costs only (no shutdown). Strategy 2: moderate shutdown. Strategy 3: minimize deaths only. Bottom grid of plots: cumulative cost (top row), daily infections (middle row) and daily shutdown strategy (bottom row).

concave

$$f\left(\frac{1}{2}(x_1 + x_2)\right) \geq \frac{1}{2}(f(x_1) + f(x_2)) = f(x'),$$

and so the periodic strategy is more expensive than a constant strategy with the same effectiveness. For example, if we take $X_t \equiv x_c$ for the whole period of $T$ days, the shutdown cost is \$80.3 billion. If we alternate between values $x_1 = 0.28$ and $x_2 = 0.09$ (which satisfy $\frac{1}{2}(f(x_1) + f(x_2)) = f(x_c)$) then the shutdown cost increases to \$90.3 billion.

# Appendix B. Shutdown is non-zero

In §2.2, we chose the shutdown effectiveness function $f(x) = x^\theta$, with $\theta < 1$. We now show that this implies that the optimal strategy (with respect to total costs) is non-zero. For simplicity, we just treat the case when we ignore excess deaths.

In a SEIR model, the final population of susceptibles $S_\infty$ satisfies [59]

$$\ln\left(\frac{S_0}{S_\infty}\right) = \mathcal{R}_0(1 - N^{-1}S_\infty).$$

It follows that $S_\infty$ is a differentiable function of $\mathcal{R}_0$: write $S_\infty = H(\mathcal{R}_0)$. Let us write $S_\infty$ for the final number of susceptibles if we adopt a modified strategy $X_t \equiv 0$, and $\tilde{S}_\infty$ for the final number if $X_t \equiv x$, where $x$ is small. Then

$$\tilde{S}_\infty - S_\infty = H(\mathcal{R}_0(x)) - H(\mathcal{R}_0) = H(\mathcal{R}_0 - x^\theta \mathcal{R}_0) - H(\mathcal{R}_0) \simeq -x^\theta \mathcal{R}_0 H'(\mathcal{R}_0).$$

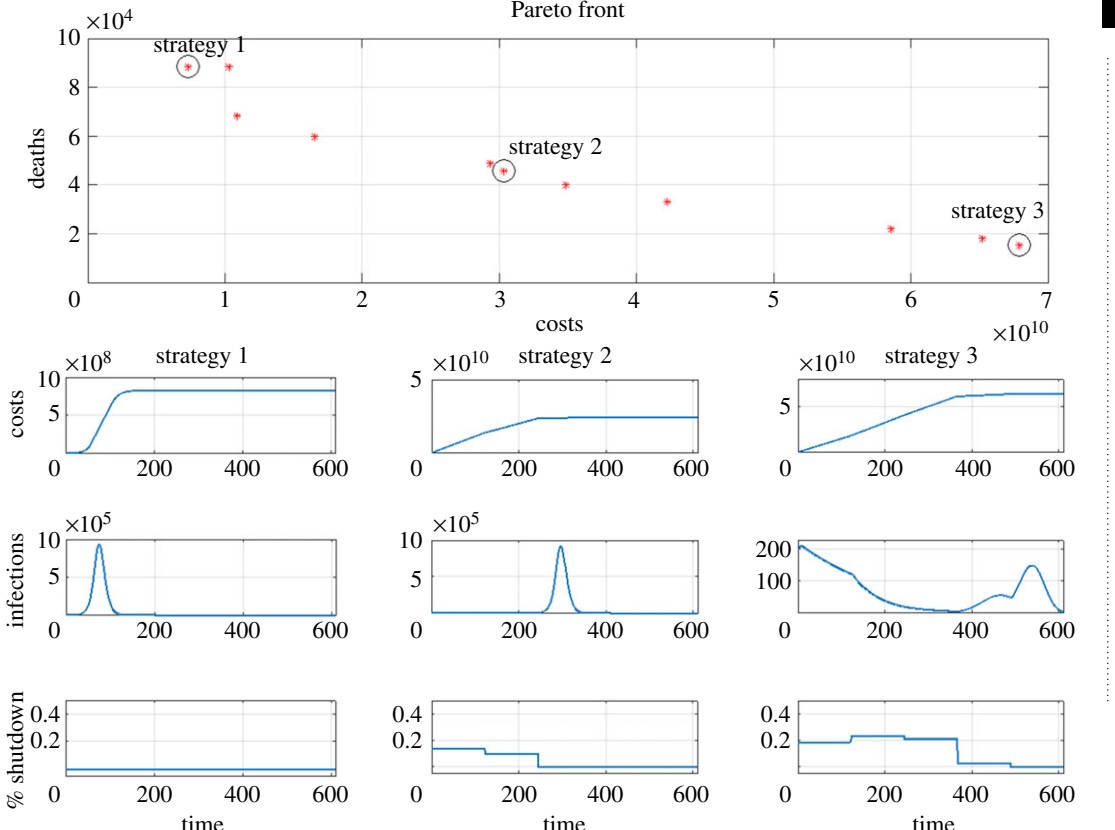

**Figure 8.** Top: Pareto plot for $\mathcal{R}_0 = 5.0$. Each asterisk corresponds to the shutdown strategy yielding the approximately minimum possible number of deaths $D_t$ at dollar cost $C^{(\text{Dol})}$ (i.e. not including the value of a lost life). The Pareto front assumes default parameter values and five shutdown intervals. Strategy 1: minimize costs only (no shutdown). Strategy 2: moderate shutdown. Strategy 3: minimize deaths only. Bottom grid of plots: cumulative cost (top row), daily infections (middle row) and daily shutdown strategy (bottom row).

Hence the modified strategy leads to a cost saving (in terms of medical and deaths costs) of

$$x^\theta \mathcal{R}_0 H'(\mathcal{R}_0) p_{\text{IFR}} V_L.$$

On the other hand, the economic cost of the modified strategy is $xg_D T$, so writing $C^{(\text{Tot})}$ and $\tilde{C}^{(\text{Tot})}$ for the total costs of the original and modified strategies, we have

$$C^{(\text{Tot})} - \tilde{C}^{(\text{Tot})} = x^\theta \mathcal{R}_0 H'(\mathcal{R}_0) p_{\text{IFR}} V_L - xg_D T,$$

which is positive for small enough $x$.

A similar analysis shows that if we consider piecewise constant strategies, the optimal strategy $X_t$ satisfies $X_t > 0$ in any time period in which the initial number of infected is greater than zero.

# Appendix C. Pareto fronts for larger $\mathcal{R}_0$

The default value $\mathcal{R}_0 = 3.3$ is appropriate for the original SARS-CoV-2 virus. Mutations have arisen, however, giving rise to more infectious variants, with the Delta variant having $\mathcal{R}_0 \approx 5$. With increasing infectiousness, the Pareto curve straightens out, as shown in figures 7 and 8.

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
