## [Peer Review File · Royal Society Open Science]

Review History

RSOS-202255.R0 (Original submission)

Review form: Reviewer 1

Is the manuscript scientifically sound in its present form?

Yes

Are the interpretations and conclusions justified by the results?

Yes

Is the language acceptable?

Yes

Do you have any ethical concerns with this paper?

Yes

Have you any concerns about statistical analyses in this paper?

Yes

Recommendation?

Accept with minor revision (please list in comments)

Comments to the Author(s)**COMMENTS**

I like the paper and I really have no objections about the methods or results.

Just recommendations. One recommendation is to make figures more readable (larger fonts and higher quality for sure). Another recommendation is about running a global sensitivity and uncertainty analysis (see Pianosi et al., 2016) that consider factor interactions rather than the one-factor-at-a-time sensitivity analysis as done by the authors; this may not reveal important nonlinearities in the model. The last recommendation is about citing recent papers that are conceptually aligned to the portfolio/Pareto optimization idea: e.g. see Chan et al. (2021) where authors run an extensive analysis on data to define the best set of interventions. I believe the authors of this submitted manuscript made the whole study theoretical/computational but I would love to see in the future how their 1D model applies to real world country data. Maybe some validations on real-world data can be done before resubmission.

For the reasons above I suggest Minor Revisions.

Pianosi et al. (2016)

Sensitivity analysis of environmental models: A systematic review with practical workflow
Environmental Modelling & Software
Volume 79, May 2016, Pages 214-232

Louis Yat Hin Chan et al. (2021)

COVID-19 non-pharmaceutical intervention portfolio effectiveness and risk communication
predominance
Scientific Reports volume 11, Article number: 10605

Review form: Reviewer 2 (Katharina Hauck)

Is the manuscript scientifically sound in its present form?

Yes

Are the interpretations and conclusions justified by the results?

Yes

Is the language acceptable?

Yes

Do you have any ethical concerns with this paper?

No

Have you any concerns about statistical analyses in this paper?

Yes

Recommendation?

Accept with minor revision (please list in comments)

Comments to the Author(s)

Dear Authors, please find my referee report attached (see Appendix A).

Decision letter (RSOS-202255.R0)

Dear Dr Tyson,

On behalf of the Editors, we are pleased to inform you that your Manuscript RSOS-202255 "Optimal shutdown strategies for COVID-19 with economic and mortality costs: BC as a case study" has been accepted for publication in Royal Society Open Science subject to minor revision in accordance with the referees' reports. Please find the referees' comments along with any feedback from the Editors below my signature.

Please submit your revised manuscript and required files (see below) no later than 7 days from today's (ie 08-Jul-2021) date. Note: the ScholarOne system will 'lock' if submission of the revision is attempted 7 or more days after the deadline. If you do not think you will be able to meet this deadline please contact the editorial office immediately.

on behalf of Professor Christine Currie (Associate Editor) and Mark Chaplain (Subject Editor)
openscience@royalsociety.org

Associate Editor Comments to Author (Professor Christine Currie):

Both referees agree that this paper is of a high standard but there are still some minor changes to be made before publication.

A few points:

The presentation of equations 2.1c and 2.1d might need some thought given the complexities introduced on the following page. The discussion may need some reconsidering.

The assumption that even asymptomatic patients will eventually be quarantined seems very optimistic and I wonder whether there might be a better way of modelling this.

It would be useful to include vaccination in the model diagram in Figure 1 to better understand how it has been modelled.

There is an author's note in Figure 4.

Reviewer comments to Author:

Reviewer: 1

Comments to the Author(s)

COMMENTS

I like the paper and I really have no objections about the methods or results.

Just recommendations. One recommendation is to make figures more readable (larger fonts and higher quality for sure). Another recommendation is about running a global sensitivity and uncertainty analysis (see Pianosi et al., 2016) that consider factor interactions rather than the one-factor-at-a-time sensitivity analysis as done by the authors; this may not reveal important non-linearities in the model. The last recommendation is about citing recent papers that are conceptually aligned to the portfolio/Pareto optimization idea: e.g. see Chan et al. (2021) where authors run an extensive analysis on data to define the best set of interventions. I believe the authors of this submitted manuscript made the whole study theoretical/computational but I would love to see in the future how their 1D model applies to real world country data. Maybe some validations on real-world data can be done before resubmission.

For the reasons above I suggest Minor Revisions.

Pianosi et al. (2016)

Sensitivity analysis of environmental models: A systematic review with practical workflow
Environmental Modelling & Software
Volume 79, May 2016, Pages 214-232

Louis Yat Hin Chan et al. (2021)

COVID-19 non-pharmaceutical intervention portfolio effectiveness and risk communication
predominance
Scientific Reports volume 11, Article number: 10605

Reviewer: 2

Comments to the Author(s)

Dear Authors, please find my referee report attached.

===PREPARING YOUR MANUSCRIPT===

===PREPARING YOUR REVISION IN SCHOLARONE===

- If you are providing image files for potential cover images, please upload these at this step, and inform the editorial office you have done so. You must hold the copyright to any image provided.
- A copy of your point-by-point response to referees and Editors. This will expedite the preparation of your proof.

- Ensure that your data access statement meets the requirements at <https://royalsociety.org/journals/authors/author-guidelines/#data>. You should ensure that you cite the dataset in your reference list. If you have deposited data etc in the Dryad repository, please only include the 'For publication' link at this stage. You should remove the 'For review' link.
- If you are requesting an article processing charge waiver, you must select the relevant waiver option (if requesting a discretionary waiver, the form should have been uploaded at Step 3 'File upload' above).
- If you have uploaded ESM files, please ensure you follow the guidance at <https://royalsociety.org/journals/authors/author-guidelines/#supplementary-material> to include a suitable title and informative caption. An example of appropriate titling and captioning may be found at https://figshare.com/articles/Table_S2_from_Is_there_a_trade-off_between_peak_performance_and_performance_breadth_across_temperatures_for_aerobic_scorpions_in_teleost_fishes_/3843624.

Author's Response to Decision Letter for (RSOS-202255.R0)

See Appendix B.

Decision letter (RSOS-202255.R1)

Dear Dr Tyson,

I am pleased to inform you that your manuscript entitled "Optimal shutdown strategies for COVID-19 with economic and mortality costs: BC as a case study" is now accepted for publication in Royal Society Open Science.

COVID-19 rapid publication process:

We are taking steps to expedite the publication of research relevant to the pandemic. If you wish, you can opt to have your paper published as soon as it is ready, rather than waiting for it to be published the scheduled Wednesday.

This means your paper will not be included in the weekly media round-up which the Society sends to journalists ahead of publication. However, it will still appear in the COVID-19 Publishing Collection which journalists will be directed to each week (<https://royalsocietypublishing.org/topic/special-collections/novel-coronavirus-outbreak>).

If you wish to have your paper considered for immediate publication, or to discuss further, please notify openscience_proofs@royalsociety.org and press@royalsociety.org when you respond to this email.

on behalf of Professor Christine Currie (Associate Editor) and Mark Chaplain (Subject Editor)
openscience@royalsociety.org

Appendix A

Referee report for RSOS-202255 Optimal shutdown strategies for COVID-19 with economic and mortality costs: BC as a case study

The paper develops an epidemiological model of SARS-CoV-2 transmission that projects the impact of reductions in economic production (due to pandemic mitigation, i.e., closures of businesses) on lost GDP, hospital treatment costs, and deaths (valued in monetary terms with Value of Statistical Life). The model is used to determine the optimal, i.e. cost minimizing, level of economic closure. It also quantifies the trade-off between economic costs due to business closures and societal costs due to deaths. It is used to estimate the Pareto optimal frontier when trading off deaths and economic costs.

The pandemic has shown us how urgently we need integrated economic-epidemiological modelling to advise policy makers on the agonizing trade-off between lives and livelihoods. This paper addresses this gap and contributes to the rapidly developing literature on integrated modelling. It stands out from other similar papers in that the epidemiological part of the model is developed with greater sophistication than the economic part. Overall, and despite its limitations, I think that the paper makes an important contribution and gets us one step closer to economic-epidemiological modelling. I have a number of reservations on the model and the way it is written up, and a few suggestions for improvements.

Economic modelling and link between epidemiological and economic model is very simplified. A limitation of the model is that the impact of closures on the economy and the link between business closures and transmission dynamics is modelled in a very simplified manner. Closures reduce (pre-pandemic? Please clarify) GDP by a proportionate amount, but there is no differentiation by economic sectors. Clearly, some contact-heavy sectors such as hospitality or travel are closed much more than other more contact light sectors because they contribute more to transmission. This is not considered in the model, but the implicit assumption is that all sectors are closed to the same degree. The model does not differentiate by the gross value added of specific sector. Having said that, the model assumes a marginal decrease in the efficacy of closure on transmission reduction, which may implicitly capture the fact that contact heavy sectors are closed before contact light sectors. Still, the differential impact on economic production across different sectors is not considered. The authors should make this limitation clear in the discussion.

Parameters of the epidemiological model

It is easy to criticize parameter assumptions considering the fast progress of research on SARS-CoV-2, and most epidemiological models are outdated within a few months. Still, I think the model has a few outdated parameter assumptions that are probably having an impact on results, and it might be worth to conduct some sensitivity analyses.

It is unclear from the description whether the costs are projected only until day 360 (when vaccinations start), or also beyond over the vaccination period. Clearly, it takes time to vaccinate the whole population and in the meantime closures need to be kept up to keep infections under control, with all associated costs. I would therefore argue that all costs should be counted until the population is vaccinated to a level where herd immunity is reached (if this is not already done). This makes vaccination administration rate, i.e., the speed by which the population is vaccinated, a potentially important parameter. Sensitivity of the findings to this assumption is not explored by the authors, and I think that would be important.

We know now that natural and vaccine-induced immunity is waning, but as far as I see the model does not assume waning immunity. Some other models of SARS-CoV-2 have shown that projections are sensitive to assumptions of waning immunity, but I am not sure whether that would actually

make such a difference. Still, the authors should mention that as a limitation in the discussion section.

Third, the assumption of $R_0=3.3$ is unrealistic considering the rapid spread of the delta variant in many countries, and there is research to suggest the delta variant has an R_0 above 5. Increasing this parameter is likely to make quite a difference to the results. This is actually demonstrated in sensitivity analyses by the authors, but it would be good to see the actual difference in projections. The model is not differentiated by age, which is a limitation considering the variation in case fatality rate and severe illness by age.

Economic impact estimate is limited

The authors should make very clear that what is estimated here is short-term GDP loss. While capital destruction may lead to a greater mid-term GDP loss (as the author mentions), there is also the possibility of a fast recovery that brings GDP back to the pre-pandemic level very quickly. There may even be over-production due to pent-up demand. The authors should mention that in the limitations. Changes to imports and exports are not considered, or in this application, the economic exchange between British Columbia and the rest of the country, and indeed the world. The impact of illness on the work force is not considered in the economic impact estimates, and neither is reduction in productivity (or transmission!) due to working from home.

Better presentation and explanation of the results.

There are relatively few results presented, and the discussion of results is short. For example, why is there no shutdown at all for strategy 1? Is that because the hospital costs are lower than the economic costs of shutdowns? If yes, please explain. In figure 4, why are there discontinuities in the pandemic curve? Is that because of stepwise reductions in shutdowns? Please explain. On page 13, it is stated that we do not see excess deaths until VSL valuations are very low. Is this because at higher valuations of deaths the model does not let infections go sufficiently high to breach hospital capacity? Please explain. It is confusing to mention in this section VSL, because in the Pareto optimization deaths are not valued in monetary terms as far as I understand? On page 15, reference is made to variation in social distancing. That is the first time as far as I can see that reference is made to variation in social distancing, which is confusing. Is that modelled? Please clarify? The modelling of increased case fatality rate when hospital occupancy breaches capacity is not shown in the results for the unmitigated scenarios, although that is a strength of the model. It would be interesting to know how many of such deaths are projected.

Minor comments

Page 6 top: It seems there is a discrepancy between figure 1 and the text: Isn't the parameter ρ_{wd} the fatality rate of those who are waiting for hospital treatment? And not of those waiting for ICU? Figure 1 suggests that for those waiting for ICU the FR is 1? Please clarify

Appendix B

RSOS-202255. Optimal shutdown strategies for COVID-19 with economic and mortality costs: BC as a case study

We wish to thank the editors and reviewers for their careful reading of our manuscript, and for giving us the opportunity to submit a revision. Below, the comments from the editors and reviewers appear in italics, and our responses appear in regular type. In the manuscript itself, revisions to the text appear in blue, with the exception of minor changes (e.g. corrections of typos).

Response to reviewers

Comments of Associate Editor

(1) The presentation of equations 2.1c and 2.1d might need some thought given the complexities introduced on the following page. The discussion may need some reconsidering.

We have modified these equations to include vaccination. The container Q is the sum of the containers introduced in equations (2.2a)-(2.f); we have explained this point in line 69.

(2) The assumption that even asymptomatic patients will eventually be quarantined seems very optimistic and I wonder whether there might be a better way of modelling this.

When we began our work, the number, and even the existence, of asymptomatic and infectious patients was unclear. Adding additional containers to handle this group would increase the complexity of the model, and the number of parameters. We think it unlikely that this modification would make much difference to the overall pattern of our results, but of course this could only be determined for certain through further work. We have added some remarks on asymptomatic patients near the end of Section 2.1 (lines 73-77).

(3) It would be useful to include vaccination in the model diagram in Figure 1 to better understand how it has been modelled.

We have modified the Figure.

(4) There is an author's note in Figure 4.

Removed.

Comments of Reviewer 1

(1) One recommendation is to make figures more readable (larger fonts and higher quality for sure).

We have improved the figures.

(2) *Another recommendation is about running a global sensitivity and uncertainty analysis (see Pianosi et al., 2016) that consider factor interactions rather than the one-factor-at-a-time sensitivity analysis as done by the authors; this may not reveal important non-linearities in the model.*

As well as the one-factor at a time parameter changes described in Table 3, our original manuscript also included results from a sensitivity analysis with the PRCC method, that includes factor interactions (Section 3.2). We originally referred to Marino (2008), and have now added a reference to Pianosi (2016) (line 248). Of course more could be done, but it a deeper analysis of the model sensitivity would involve a very substantial amount of additional work, and would not change our main conclusions.

(3) *The last recommendation is about citing recent papers that are conceptually aligned to the portfolio/Pareto optimization idea: e.g. see Chan et al. (2021) where authors run an extensive analysis on data to define the best set of interventions.*

We have added a reference to the Chan (2021) paper (lines 87-90), and a reference to another recent paper that we found which looked at costs-vs-deaths Pareto fronts (Wulkow 2021, lines 363-365).

(4) *I believe the authors of this submitted manuscript made the whole study theoretical/computational but I would love to see in the future how their 1D model applies to real world country data. Maybe some validations on real-world data can be done before resubmission.*

We have added some text to the conclusions outlining the value of our theoretical approach (lines 452-457). We also provide references to more data-driven COVID-19 modelling papers that obtain similar results, thus validating our work. The fact that our relatively simple, theoretical model produces results that are consistent with considerably more complex ones suggests that our model captures the essential elements of the coupled disease+economics system, and is thus useful for development of an intuitive understanding of the system.

(5) *Suggested additional references.*

ref1) *Pianosi et al. (2016)*

Sensitivity analysis of environmental models: A systematic review with practical workflow Environmental Modelling & Software. Volume 79, May 2016, Pages 214-232

ref2) *Louis Yat Hin Chan et al. (2021)*

COVID-19 non-pharmaceutical intervention portfolio effectiveness and risk communication predominance. Scientific Reports volume 11, Article number: 10605

These references have been added.

Comments of Reviewer 2

(1) *A limitation of the model is that the impact of closures on the economy and the link between business closures and transmission dynamics is modelled in a very simplified manner. The authors should make this limitation clear in the discussion.*

We considered a very simplified model, both to try to focus on the essentials and also because a more realistic model would involve many unknown parameters. We added a few more lines to emphasise this point at the start of Section 2.2 (lines 84-89), and provide a reference to another paper that takes a similar approach (Chao 2021).

(2) *It is easy to criticize parameter assumptions considering the fast progress of research on SARS-CoV-2, and most epidemiological models are outdated within a few months. Still, I think the model has a few outdated parameter assumptions that are probably having an impact on results, and it might be worth to conduct some sensitivity analyses.*

We did present a sensitivity analysis in our original manuscript - see Section 3.2. We added some remarks on the difficulty of modelling an evolving disease at the end of Section 1 (lines 48-51). We also considered values of R_0 much larger than those originally looked at - see the additions to Table 3 and the remarks at the end of Section 3.2 (lines 267-270), as well as the new Pareto front plots for $R_0=4$ and 5 in the new Appendix C (pages 22-24).

(3) *It is unclear from the description whether the costs are projected only until day 360 (when vaccinations start), or also beyond over the vaccination period. Clearly, it takes time to vaccinate the whole population and in the meantime closures need to be kept up to keep infections under control, with all associated costs. I would therefore argue that all costs should be counted until the population is vaccinated to a level where herd immunity is reached (if this is not already done).*

Our model does run until the epidemic is ended by vaccination; we added a remark at lines 139-140 to emphasize this point.

(4) *This makes vaccination administration rate, i.e., the speed by which the population is vaccinated, a potentially important parameter. Sensitivity of the findings to this assumption is not explored by the authors, and I think that would be important.*

We have tried varying the vaccination rate -- naturally it makes a difference to costs but not to the basic pattern of how the epidemic is controlled. We added a line to Table 3, and some comments at the end of Section 3.2 (lines 278-280).

(5) *We know now that natural and vaccine-induced immunity is waning, but as far as I see the model does not assume waning immunity. Some other models of SARS-CoV-2 have shown that projections are sensitive to assumptions of waning immunity, but I am not sure whether that*

would actually make such a difference. Still, the authors should mention that as a limitation in the discussion section.

We have added some remarks on this at the end of Section 4.4 (lines 442-448)

(6) *Third, the assumption of $R_0=3.3$ is unrealistic considering the rapid spread of the delta variant in many countries, and there is research to suggest the delta variant has an R_0 above 5. Increasing this parameter is likely to make quite a difference to the results.*

We have now looked at much larger R_0 - see the remarks on point (2) above.

(7) *The model is not differentiated by age, which is a limitation considering the variation in case fatality rate and severe illness by age.*

We do make some remarks in Section 4.2, and have added a note at the end of that section (lines 387-389).

(8) *Economic impact estimate is limited. The authors should make very clear that was is estimated here is short-term GDP loss.*

We consider long term effects to be beyond the scope of this paper, and have added some comments at the end of Section 2.2, to emphasize this point (lines 149-156). We have also updated our discussion of capital losses in the Discussion (lines 412-419), based on currently available data. There is still considerable uncertainty in this area, however, and so we leave this as a matter for future study.

(9) *Why is there no shutdown at all for strategy 1?*

Strategy 1 (Figure 6) is not actually on the true Pareto front - the MATLAB Pareto code was not able to find a better point with a very small but non-zero shutdown. We have added some warnings about the computed Pareto front in Section 3.3 (lines 294-297), and in particular discuss its failure with respect to Strategy 1 (lines 304-306). We have also modified the caption to Figure 6.

(10) *In figure 4, why are there discontinuities in the pandemic curve? Is that because of stepwise reductions in shutdowns? Please explain.*

The shutdown strategy was chosen to be piecewise constant, so it has discontinuities. The number of infectious patients is continuous, but its derivative does have discontinuities. We have given the reasons for this behaviour in the second paragraph of section 3.1.2 (lines 220-224) - a change in shutdown leads almost immediately to a change in the number of infections.

(11) *On page 13, it is stated that we do not see excess deaths until VSL valuations are very low. Is this because at higher valuations of deaths the model does not let infections go sufficiently high to breach hospital capacity? Please explain. It is confusing to mention in this section VSL, because in the Pareto optimization deaths are not valued in monetary terms as far as I understand?*

We have reorganised the paper to indicate that the sensitivity analysis, as well as the various scenarios listed in Table 3, was done for the simple cost optimization, not the Pareto optimization. As the reviewer has guessed, except for very low VSL, the epidemic is controlled well enough to avoid medical overload: We added a remark in Section 3.1.2 (blue text, lines 225-233).

(12) *On page 15, reference is made to variation in social distancing. That is the first time as far as I can see that reference is made to variation in social distancing, which is confusing. Is that modelled? Please clarify?*

This was an error, and we have removed the reference.

(13) *The modelling of increased case fatality rate when hospital occupancy breaches capacity is not shown in the results for the unmitigated scenarios, although that is a strength of the model. It would be interesting to know how many of such deaths are projected.*

We do give figures for excess deaths for some scenarios in Table 3; and we have added text on excess deaths for Scenario 1 in Section 3.3 (lines 303-306).

(14) *Page 6 top: It seems there is a discrepancy between figure 1 and the text: Isn't the parameter ρ the fatality rate of those who are waiting for hospital treatment? And not of those waiting for ICU? Figure 1 suggests that for those waiting for ICU the FR is 1? |*

We have corrected the Figure. Also we now explain in the caption that the container W_U is 'instantaneous'.